# Cas9 exo-endonuclease eliminates chromosomal translocations during genome editing

Jianhang Yin[1,4], Rusen Lu[1,4], Changchang Xin[1,4], Yuhong Wang[1], Xinyu Ling[2], Dong Li[1], Weiwei Zhang[1], Mengzhu Liu[1], Wutao Xie[1], Lingyun Kong[1], Wen Si[1], Ping Wei[1], Bingbing Xiao[3], Hsiang-Ying Lee [1], Tao Liu [2] & Jiazhi Hu [1✉]

The mechanism underlying unwanted structural variations induced by CRISPR-Cas9 remains poorly understood, and no effective strategy is available to inhibit the generation of these byproducts. Here we find that the generation of a high level of translocations is dependent on repeated cleavage at the Cas9-targeting sites. Therefore, we employ a strategy in which Cas9 is fused with optimized TREX2 to generate Cas9TX, a Cas9 exo-endonuclease, which prevents perfect DNA repair and thereby avoids repeated cleavage. In comparison with CRISPR-Cas9, CRISPR-Cas9TX greatly suppressed translocation levels and enhanced the editing efficiency of single-site editing. The number of large deletions associated with Cas9TX was also reduced to very low level. The application of CRISPR-Cas9TX for multiplex gene editing in chimeric antigen receptor T cells nearly eliminated deleterious chromosomal translocations. We report the mechanism underlying translocations induced by Cas9, and propose a general strategy for reducing chromosomal abnormalities induced by CRISPR-RNA-guided endonucleases.

[1] The MOE Key Laboratory of Cell Proliferation and Differentiation, School of Life Sciences, Center for Life Sciences, Genome Editing Research Center, Peking University, 100871 Beijing, China. [2] State Key Laboratory of Natural and Biomimetic Drugs, School of Pharmaceutical Sciences, Peking University, 100191 Beijing, China. [3] Department of Obstetrics and Gynecology, Peking University First Hospital, 100034 Beijing, China. [4] These authors contributed equally: Jianhang Yin, Rusen Lu, Changchang Xin. ✉email: hujz@pku.edu.cn

CRISPR-Cas9 generates DNA double-stranded breaks (DSBs) to achieve efficient gene disruptions for the purpose of curing genetic disorders[1–5]. However, in addition to insertions and deletions (indels), *Streptococcus pyogenes* Cas9 (*Sp*Cas9) nuclease has been reported to lead to unwanted structural variations, including chromosomal translocations, large deletions, and other complex rearrangements[6–10]. Among these chromosomal abnormalities, chromosomal translocations receive particular attention. Chromosomal translocations pose a great threat to genome integrity and are the most prevalent type of DNA mutations in hematological malignancies[11,12]. Specifically, oncogenic translocations are frequently observed in patients with T cell acute lymphoblastic leukemia, and more than one-third of observed translocations involve the *TCR* (*TRAC* and *TRBC*) loci[13,14]. However, an effective method for reducing chromosomal translocations induced by *Sp*Cas9 and similar Cas nucleases has not been established.

The escaped broken ends of DSBs from target sites, off-target sites, and genome-wide general DSBs within a cell may fuse to form chromosomal translocations, as previously reported in single-gene-edited mouse embryonic stem cells (mESCs), neuron stem cells, and other human cell lines[10,15–17]. Chromosomal translocations occur at an estimated frequency of one in 100–1000 human cells with *Sp*Cas9-induced DSBs or other DSBs[18–20]. Furthermore, the concern regarding translocations is exacerbated in multiplex gene editing[10,15,16,21]. In this context, several genes involved in alloreactivity (*TRAC, TRBC*, and *B2M*) and immunosuppression (*PDCD1*) are simultaneously disrupted by *Sp*Cas9 nuclease to optimize chimeric antigen receptor (CAR) T cell therapy and T cell receptor (TCR) T cell therapy. A recent clinical study reported that engineered CAR T cells with translocations among these target sites can exist in vivo for up to 170 days or more after reinfusion into patients[10].

In this work, we identify a large number of chromosomal translocations in T cells targeted at the *TRAC, TRBC*, and *PDCD1* genes on day 3 post-editing, and some of them are still present on days 7 and 14. We find that the high level of translocations resulted from repeated cleavage at the target loci of CRISPR-Cas9. In order to reduce the number of translocations, we propose a general strategy in which an exonuclease domain is fused to *Sp*Cas9 to generate an exo-endonuclease form and thus prevent repeated cleavage during multiplex genome editing. Here, we fuse *Sp*Cas9 with optimized TREX2 to generate a Cas9 exo-endonuclease termed Cas9TX. Cas9TX greatly reduces the occurrence of chromosomal abnormalities in both engineered T cells and other tested cells with single-site gene editing. Notably, the editing efficiency at many target loci is enhanced by Cas9TX in comparison with *Sp*Cas9. Our findings provide a general method for the application of engineered sequence-specific nucleases to reduce the tendency for chromosomal abnormalities.

## Results

### Substantial chromosomal translocations in gene-edited T cells.
To sensitively detect translocations in CRISPR-Cas9-edited cells, we adapted the previously described primer extension-mediated sequencing (PEM-seq) method for translocation capture. Generally, a bait primer is placed at one of the broken ends of the target sites to generate a PEM-seq library. PEM-seq captures insertions/deletions (indels) between the two broken ends of the target sites, as well as translocations between the bait broken ends and other DSB ends (Supplementary Fig. 1a; refs. [9,22,23].). In the PEM-seq data, the ratio of indels to total sequencing junctions is defined as the editing efficiency, while the ratio of translocations to indels plus translocations is defined as the percentage of translocations.

For CAR T generation, T cells were enriched from human core blood and activated by anti-CD3/CD28 for 3 days before CRISPR-Cas9 editing at the *TRAC, TRBC*, and *PDCD1* genes as described in clinical protocol NCT03399448[10]. Purified *Sp*Cas9 protein was mixed with three single guide RNAs (sgRNAs, at 1:1:1) to allow it to be delivered in ribonucleoprotein complexes (RNPs). The editing efficiency and translocations in T cells were monitored at 3-, 7-, and 14-days post-transfection (Fig. 1a and Supplementary Fig. 1b). The editing efficiency of CRISPR-Cas9 was ~47.7% for *TRAC*, 52.9% for *TRBC*, and 64.2% for *PDCD1* at 3-days post-transfection, as determined by PEM-seq (Fig. 1b, Supplementary Table 1). Of note, the variation in editing efficiency of these replicates may have been due to the use of different batches of purified Cas9 or core blood for the experiments. Moreover, the sgRNA for *TRBC* has two bona fide recognition sites ~9.4 kb away from each other within the *TRBC* gene, and we combined them for the subsequent analysis. As the culture time was increased to 7 or 14 days, the number of cells containing indels gradually decreased, and this effect was probably due to retardation of the growth of some of the edited cells (Fig. 1b).

To capture all of the reciprocal translocations among *TRAC, TRBC*, and *PDCD1*, we generated PEM-seq libraries with a bait primer for each target site individually. The percentages of identified translocations among the three targeted genes varied from 0.23 to 1.28% at 3-days post-transfection (Fig. 1c, d). For example, the level of chromosomal translocations between *TRAC* and *TRBC* was 0.23% when *TRAC* was used as bait, whereas it was 0.54% when *TRBC* was used as bait (Fig. 1c, d). These translocations were further validated by nested PCR with primers spanning the translocation junctions (Supplementary Fig. 1c). We also identified an off-target site for *Sp*Cas9:*TRAC* lying at the subtelomere region of chromosome X as previously reported[10], with editing efficiency ranging from 1.0 to 1.9% (Supplementary Fig. 1d). The levels of translocations between *TRAC* off-target sites and the three targeted genes were much lower than those among the on-target sites, ranging from 0.001 to 0.003% (Fig. 1c). The translocation levels based on the PEM-seq analysis at 7- and 14-days post-transfection showed a decreasing trend (Fig. 1d), in line with a previous report[10]. However, the translocation levels remained greater than 0.1% at 14-days post-transfection (Fig. 1d and Supplementary Fig. 1e), implying that at least $1 \times 10^5$ of translocation-containing T cells would be reinfused into patients after 7–14 days if $\sim 1 \times 10^8$ engineered T cells are initially transferred.

In addition to *Sp*Cas9-induced DSBs at on- and off-target sites, genome-wide occasional DSBs occurring simultaneously with *Sp*Cas9-induced DSBs may also fuse with on-target DSBs to form chromosomal translocations[24]. Hereafter, these products are referred to as general translocations to distinguish them from *Sp*Cas9-dependent off-target translocations. Although general translocations occur at low levels and may not recur in different batches of CRISPR-Cas9-edited cells, they can also be captured by PEM-seq. General translocations were found to be distributed widely across the entire genome; the percentages of general translocations ranged from 0.60 to 2.04% for the three target sites at 3-days post-transfection (Fig. 1e and Supplementary Fig. 1f). The level of general translocations decreased with increasing culture time, similar to that of *Sp*Cas9-induced DSBs (Fig. 1d, e). Among the 5,343 identified general translocations, ~200 genes involved in various cancer pathways were observed to fuse with the *TRAC, TRBC*, or *PDCD1* gene (Fig. 1f). In addition to translocations detected in human T cells following multiplex genome editing, general translocations occurred at levels ranging from 1.0 to 2.4% in single-gene target sites within *Cep290, Hba, c-Myc*, or *Wrap53* loci in mESCs, with editing efficiency ranging

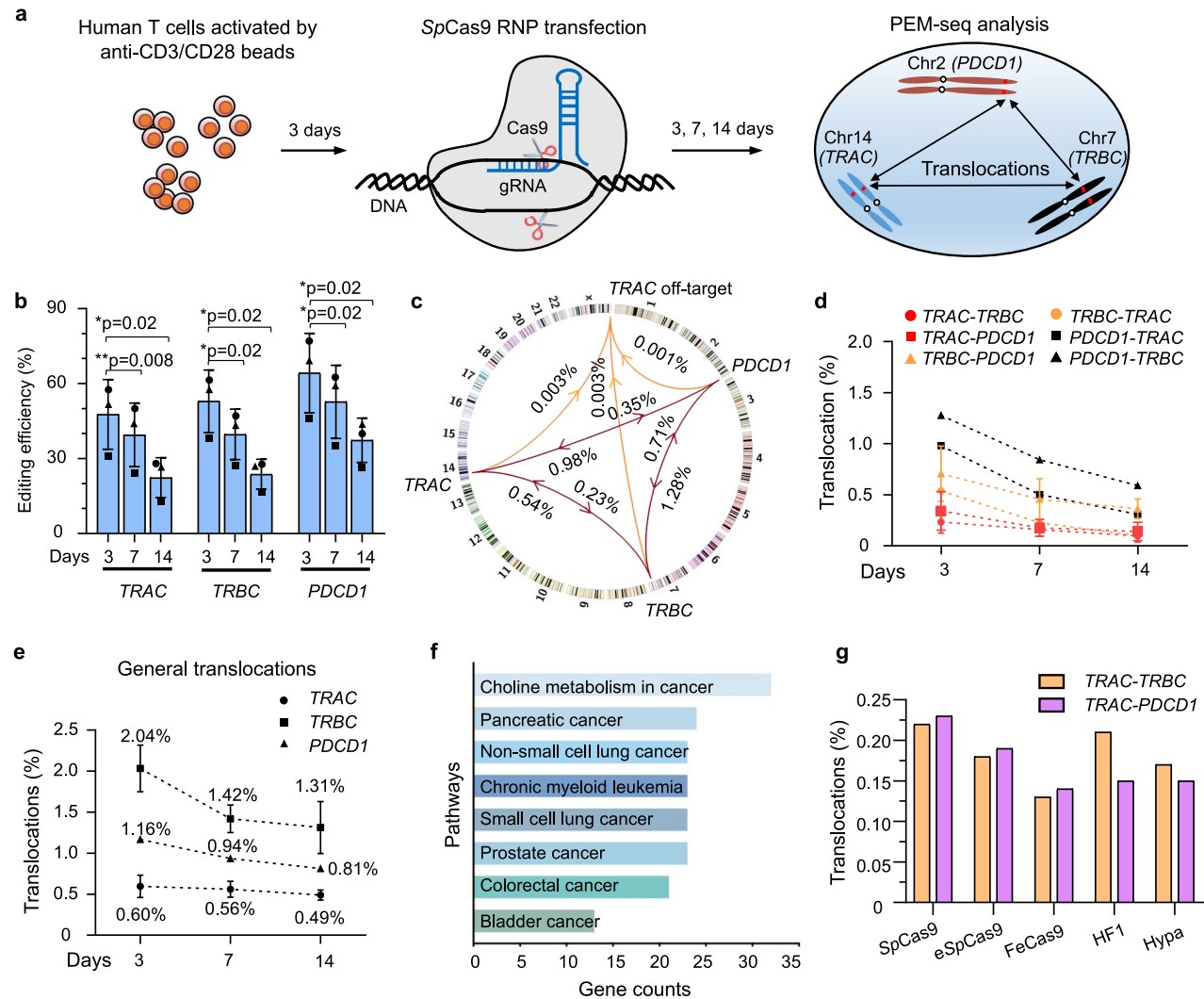

**Fig. 1 Translocations during *Sp*Cas9-mediated multiplex gene editing in human T cells. a** Schematics for assessing chromosomal translocations during multiplex gene editing by *Sp*Cas9 RNPs in human T cells. Chr represents Chromosome. **b** Editing efficiency for *TRAC*, *TRBC*, and *PDCD1* in human T cells at 3-, 7-, and 14-days post-transfection detected by PEM-seq. Mean ± SD from three biological replicates, represented by "circle", "triangle", and "square", respectively. Paired two-tailed *t*-test, *p < 0.05 and **p < 0.01. **c** Circos plot indicating translocations among *TRAC*, *TRBC*, *PDCD1*, and the *TRAC* off-target in human T cells at 3-days post-transfection detected by PEM-seq. Chromosomes are shown with centromere to telomere in the clockwise direction. The averages of three replicates are shown on each line, and the arrows indicate the orientation of bait to prey. **d** Percentages for translocations among *TRAC*, *TRBC*, and *PDCD1* detected by PEM-seq in human T cells at 3-, 7-, and 14-days post-transfection. The translocations were categorized as bait-prey, e.g., *TRAC-TRBC* represented the translocations between *TRAC* and *TRBC* cloned with a bait primer located at *TRAC*. Mean ± SD from three biological replicates. **e** Percentages for general translocations in human T cells cloned from the indicated loci at 3-, 7-, and 14-days post-transfection detected by PEM-seq. Mean ± SD from three biological replicates. Mean values are marked above each data point. **f** Gene annotation for total translocations from *TRAC*, *TRBC*, and *PDCD1* in human T cells at 3-days post-transfection using the KEGG feature of Enrichr (maayanlab.cloud/Enrichr/). The horizontal axis indicates the gene numbers in the indicated pathway. **g** Percentages for the indicated translocations induced by *Sp*Cas9 and indicated variants detected by PEM-seq in human T cells cloned from *TRAC* at 3-days post-transfection, N = 1. Source data are provided as a Source Data file.

from 11.6 to 15.7% (Supplementary Fig. 1g, h). These results suggest that chromosomal translocations are a universal feature of genome editing.

**High-fidelity *Sp*Cas9 variants cannot suppress translocations.** High-fidelity *Sp*Cas9 variants have been developed to improve the editing specificity of CRISPR-Cas9 editing. In this context, we tested whether high-fidelity *Sp*Cas9 variants e*Sp*Cas9, FeCas9, HF1, and Hypa suppressed translocations in edited T cells[9,25–27]. We purified these *Sp*Cas9 variants and mixed them with *TRAC*, *TRBC*, and *PDCD1* sgRNAs to transfect T cells via RNP delivery (Supplementary Fig. 1b). Finally, the genomic DNA from edited cells was subjected to PEM-seq analysis. The editing efficiency of

the *Sp*Cas9 variants was comparable to that of *Sp*Cas9 at the *TRAC* site, but the abovementioned *TRAC* off-target was not detected in these *Sp*Cas9 variant libraries at 3-days post-transfection (Supplementary Fig. 1i, j), indicating that the *Sp*Cas9 variants had higher editing specificity. However, the levels of translocations in these *Sp*Cas9 variant libraries were also comparable to that of the *Sp*Cas9 library, implying that high-fidelity *Sp*Cas9 variants are not able to suppress chromosomal translocations (Fig. 1g, Supplementary Fig. 1k, l). Similar findings were obtained when *TRBC* and *PDCD1* were used as bait, except that the editing efficiency of all of the high-fidelity variants was slightly lower than that of *Sp*Cas9 at the *PDCD1* locus (Supplementary Fig. 1i, k, and l).

**Repeated cleavage at target-analogous sites by CRISPR-Cas9**. To suppress translocations during CRISPR-Cas9-targeting, the interaction intensity and break frequency of the DSBs involved in the translocation must be reduced[28]. Since the interaction intensity between two given DSBs is relatively fixed, we focused on determining the frequency of DSBs induced by CRISPR-Cas9. The repair outcomes after CRISPR-Cas9 cleavage contain perfect rejoinings, indels, and translocations, among which perfect rejoinings can be repeatedly cleaved by CRISPR-Cas9 to increase the DSB frequency (Fig. 2a). However, repeated cleavage is invisible at the target sites because perfect rejoinings are indistinguishable from the target sequences. Alternatively, we examined the distribution of translocation junctions between *TRAC* and its off-target to verify potential repeated cleavage of CRISPR-Cas9. The *TRAC* off-target harbors four mutations in the body of the sgRNA and can generate two types of translocation products when the *TRAC* PAM-distal end is used as bait: untargetable products lacking a protospacer adjacent motif (PAM) formed by translocations between two PAM-distal ends and retargetable products formed by swapping the PAMs (Fig. 2b and Supplementary Fig. 2a).

We synthesized both untargetable and retargetable oligos and performed in vitro digestion assays with *Sp*Cas9:*TRAC*. The retargetable oligo was indeed targetable and could be cleaved by CRISPR-Cas9 as efficiently as the *TRAC* on-target site, while the untargetable product was hardly cleaved (Fig. 2c). Consistently, the translocation junctions identified by PEM-seq showed a junction bias; the abundance of the remaining retargetable products was only a quarter of that of the untargetable products in edited T cells (Fig. 2d). In theory, the two broken ends from the same DSB have an equal chance to form translocation with a third broken end (Supplementary Fig. 2b). In this context, junction bias (more than 1) means that the retargetable products may be targeted more than once by CRISPR/Cas9 in vivo. This *TRAC* off-target showed a similar junction bias in *Sp*Cas9:*TRAC*-edited HEK293T cells. An additional *TRAC* off-target site even exhibited a junction bias of 2 (Fig. 2c). Similarly, the newly identified *TRBC* and *PDCD1* off-targets also showed a junction bias of 2.3 to 3.0 in HEK293T cells (Fig. 2e).

Furthermore, we reanalyzed published data from HEK293T cells with the PEM-seq pipeline[9]. Junction bias was widely observed at the off-target sites of *DNMT1-1* and *C-MYC1*, as well as two sites within *RAG1* genes (*RAG1A* and *RAG1B*), in HEK293T cells; similar findings were also obtained at *RAG1A* off-target sites in HCT116, U2OS, and K562 cells (Fig. 2f and Supplementary Fig. 2c, and Supplementary Tables 2 and 3). The highest level of junction bias reached approximately 5.6 with a total of 490 junctions at the *RAG1A* site in HEK293T cells (Fig. 2f and Supplementary Table 2). To exclude the possibility that *Sp*Cas9 residence at broken ends[29] led to the observed junction bias, we generated new PEM-seq libraries from the other PAM-distal end at the *RAG1A* site in HEK293T cells (Supplementary Fig. 2d, left panels). Inverted biases were obtained from the PEM-seq libraries with reverse bait primers due to the switch of untargetable and retargetable products (Supplementary Fig. 2d, right panel), indicating that the junction bias was independent of *Sp*Cas9 residence on the cleaved ends. Collectively, these findings confirm repeated cleavage of target sites by CRISPR-Cas9 in various cell types.

**Repeated cleavage leads to an enhanced level of translocations**. To test the hypothesis that repeated cleavage promotes the formation of chromosomal translocations, we performed CRISPR-Cas9 editing and PEM-seq analysis in G1-arrested cells, in which the processing of broken ends is restricted, to facilitate the formation of perfectly repaired products[30]. To this end, we arrested K562 cells in the G1 phase with 5 µM CDK inhibitor palbociclib for 36 h prior to transfection with CRISPR-Cas9-targeting *HBA1* or two sites within the *C-MYC* genes (*C-MYC1* and *C-MYC2*) (Supplementary Fig. 2e). In comparison with cycling cells, the number of deletions in G1 cells was decreased at all three sites, verifying the lower level of end processing in G1-arrested cells (Supplementary Fig. 2f). The junction biases for the three *HBA1* off-targets were 1.6, 0.8, and 1.0 in the cycling K562 cells, but they were significantly increased to 2.3, 4.6, and 11.3 in the G1-arrested cells (Fig. 2g), implying robust repeated cleavage by CRISPR-Cas9 in the G1 phase. The translocation levels between *HBA1* and the three off-target sites in the G1-arrested cells were 6.0, 64.0, and 23.0 times those measured in the cycling cells, although their editing efficiency was lower (Fig. 2g and Supplementary Fig. 2g). Similar findings were obtained at the two *C-MYC* sites in the G1-arrested cells (Supplementary Fig. 2f-h).

To further verify the findings in G1-arrested cells, we designed a repeated cleavage system by adding a second sgRNA that targets the main editing products from the first round to promote second-round cleavage (Fig. 2h, top panel). We found that 60.9% of Cas9:*TP53*-sg1 editing products were single T insertions (Fig. 2h, left panel). We generated a *TP53*-sg2 sgRNA to specifically target the T insertions, and the combo of *TP53*-sg1 and *TP53*-sg2 induced at least two rounds of Cas9 cleavage. The number of T insertions was significantly reduced in Cas9:*TP53*-sg1&2-edited cells (Fig. 2h, left panel). To examine the formation of translocations, we combined *TP53* sgRNAs with either *C-MYC1* or *C-MYC2* sgRNA and identified translocations between *TP53* and *C-MYC* by PEM-seq. Cas9:*TP53*-sg1&2 induced significantly increased translocations with both *C-MYC1* (4.0% vs 1.0%) and *C-MYC2* (1.2% vs 0.5%) in comparison with Cas9:*TP53*-sg1, although their editing efficiency was similar (Fig. 2h).

We also analyzed dynamics in junction bias and translocations at earlier time points following CRISPR-Cas9 transfection. For this purpose, we performed Cas9:*C-MYC2* editing in K562 cells and then harvested unsorted cells for PEM-seq analysis beginning at 6 h post-transfection (Fig. 2i, top panel). The editing efficiency increased gradually during the first 30 h and peaked at 48 h (Fig. 2i, bottom left), indicating a clear accumulation of edited cells. The junction bias at the *C-MYC2* off-target was increased after the first 18 h, slightly neutralized at 24 h, and consistent at later time points (Fig. 2i, bottom middle). Moreover, the percentages of translocations between *C-MYC2* and its off-target were also increased in the first 30 h post-transfection, while slightly reduced at 48 h, possibly due to an increased proportion of cells with low junction bias (Fig. 2i, bottom right). Similar findings were also obtained in Cas9:*C-MYC2*-edited HEK293T cells (Supplementary Fig. 2i).

Collectively, these findings imply that repeated cleavage represented by junction bias can promote chromosomal translocations.

**Cas9 exo-endonuclease suppresses repeated cleavage and translocations**. To reduce chromosomal translocations, we fused *Sp*Cas9 with an exonuclease to suppress repeated cleavage by promoting end processing and imperfect rejoinings. To this end, we fused *Sp*Cas9 with several common exonucleases or activity-reserved truncation forms at the C terminus with a GGGGS linker, including three 3′-to-5′ exonucleases: truncated TREX1 (ΔTREX1), TREX2, and ΔMRE11, and four 5′-to-3′ exonucleases: ΔCTIP, ΔEXO1, ΔARTEMIS, and T5 (Fig. 3a). Plasmids containing these Cas9 exo-endonucleases and *C-MYC2* sgRNA were transfected into HEK293T cells, and genomic DNA was harvested

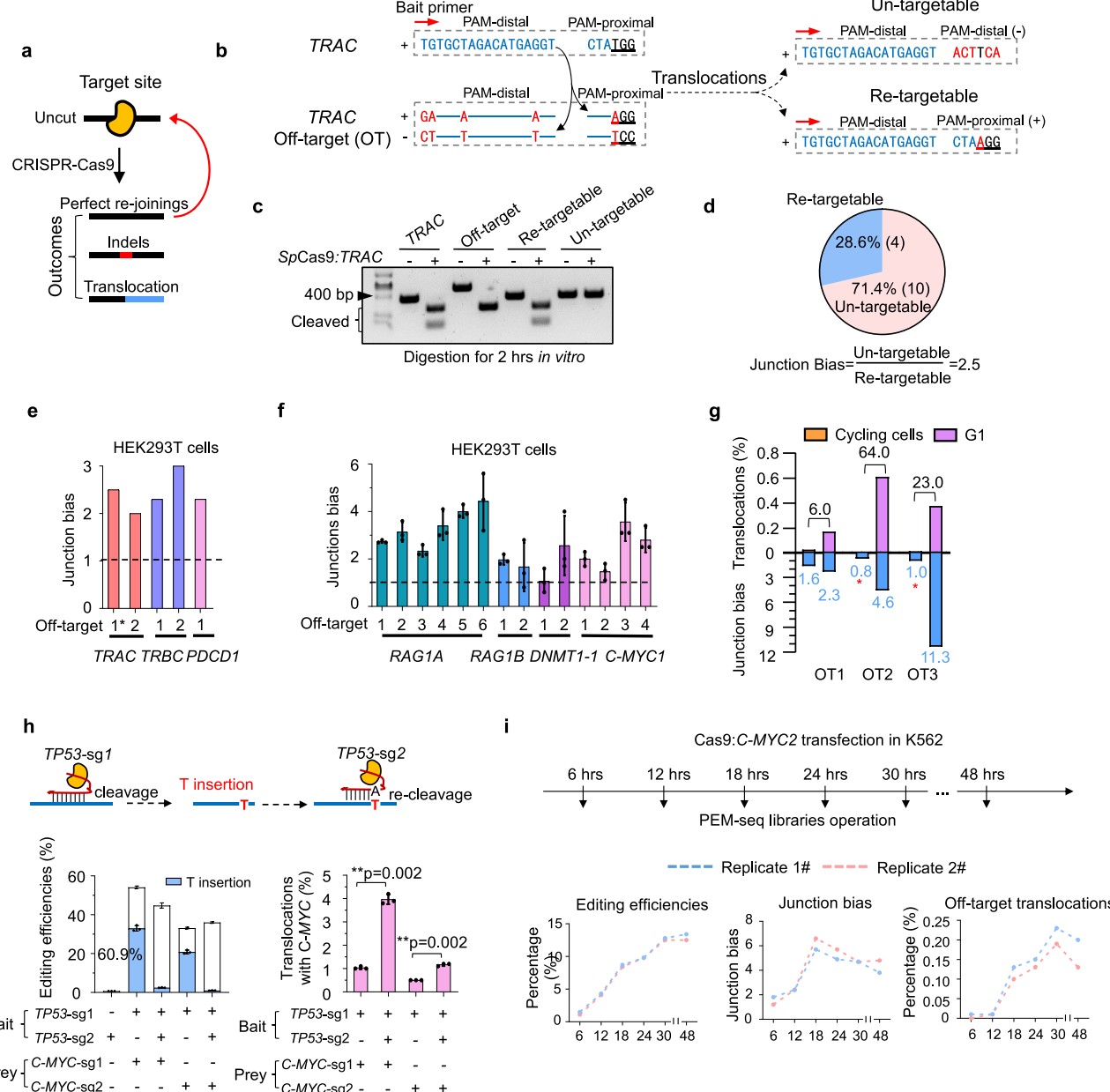

**Fig. 2 Enhancing junction bias by repeated cleavage of CRISPR-Cas9. a** Schematics for editing outcomes and repeated cleavage of CRISPR-Cas9. The perfect rejoinings and some mildly mutated repaired products can be repeatedly cleaved by CRISPR-Cas9. **b** DNA sequences of potential translocation products between *TRAC* on- and off-target sites. Red letters indicate mismatches between *TRAC* on- and off-target sites. By performing PEM-seq using the indicated bait primer (red arrow) at the *TRAC* PAM-distal end, two types of translocation products may form: "re-targetable" and "un-targetable". "+" or "−" indicates the derivation for the *TRAC* off-target strand. **c** In vitro digestion for *TRAC* on-target site, off-target site, and translocation products shown in panel **b** by *Sp*Cas9:*TRAC*. The black arrow indicates the 400 bp DNA marker. Cleaved DNA is indicated by the black bracket. **d** Percentages for "re-targetable" and "un-targetable" translocation products in human T cells cloned from the *TRAC* on-target site by PEM-seq. Junction numbers are labeled in the brackets. **e** Junction bias at the off-targets of *TRAC, TRBC,* and *PDCD1* in HEK293T cells detected by PEM-seq. The dashed line indicates the ratio = 1, $N = 1$. **f** Junction bias at the off-targets of indicated loci in HEK293T cells detected by PEM-seq. Mean ± SD from three biological replicates. The dashed line indicates the ratio = 1. The numbers on the x-axis indicate the off-target order from strong to weak. **g** Percentage for off-target translocations and junction bias at three off-target sites (OT1 to OT3) from the *HBA1* locus in K562 cells. Fold changes and junction bias are marked on the bar. Off-target sites with too few junctions are highlighted by * in red, $N = 1$ for three off-targets. **h** Schematics for two rounds of cleavage at *TP53* locus in HEK293T cells using Cas9:*TP53*-sg1 and Cas9:*TP53*-sg2. Editing efficiency and percentage of translocations for indicated treatments were detected by PEM-seq. The blue bar indicates the editing products with a T insertion. Mean ± SD from three replicates. Two-tailed t-test, *$p < 0.05$ and **$p < 0.01$. Note that 60.9% is normalized to total editing events. **i** Editing efficiency, junction bias at off-targets, and percentages for off-target translocations at *C-MYC2* locus at indicated time points in K562 cells detected by PEM-seq, $N = 2$. Source data are provided as a Source Data file.

72 h later for PEM-seq analysis. Among the tested Cas9 exo-endonucleases, Cas9-TREX2, Cas9-ΔTREX1, and Cas9-T5 exhibited higher editing efficiency, as well as correspondingly lower levels of junction bias and fewer translocations, in comparison with Cas9 (Fig. 3b, c, and Supplementary Table 4). Given that Cas9-TREX2 performed as well as Cas9-ΔTREX1, Cas9-TREX1, and Cas9-TREX2 outperformed Cas9-T5, and TREX2 has been better characterized in previous reports[31–35], we decided to use fused Cas9 and TREX2 (Cas9X2) in our subsequent experiments. We also fused Cas9 with nuclease-dead TREX2 containing an H188A mutation[36] to generate Cas9X2d.

For Cas9:*HBA1* editing in HEK293T cells, more deletions were detected in the Cas9X2 libraries in comparison with the *Sp*Cas9 libraries, suggesting that fusion with TREX2 enhanced end processing (Fig. 3d). The editing efficiency of Cas9X2 was higher than that of *Sp*Cas9 at the *HBA1* site (71.4% vs 50.5%), while Cas9X2d showed the lowest editing efficiency (Fig. 3d), supporting the notion that the processing of *Sp*Cas9-induced broken ends by TREX2 enhances genome editing[34]. Moreover, the junction bias at the three *HBA1* off-target sites was decreased with Cas9X2d in comparison with *Sp*Cas9, and Cas9X2 nearly eliminated the junction bias at all three off-target sites (Fig. 3e and Supplementary Fig. 3a). Accordingly, the levels of both off-target and general translocations were reduced by Cas9X2 and Cas9X2d, in the following order: Cas9X2 ≪ Cas9X2d < *Sp*Cas9 (Fig. 3f, g). The translocation level at the second off-target site of *HBA1* was decreased to less than 0.005% with Cas9X2 (Fig. 3f). Similar findings were obtained at the identified off-target site of *C-MYC2* (Supplementary Fig. 3b-e). We also tested eight other sites in HEK293T cells and obtained similar findings, except that the editing efficiency of Cas9X2 was generally higher, but not significantly higher, than that of *Sp*Cas9 (Fig. 3h-k, Supplementary Table 5). In addition, Cas9X2 showed a strong tendency to eliminate translocations (Fig. 3j, k); reduced levels of off-target translocations (7.7–7.2 times) and general translocations (2.7–14.3 times) were detected with Cas9X2 in comparison with *Sp*Cas9 for all target sites (Supplementary Fig. 3f, g). TREX2 has been co-expressed with Cas9 to enhance editing efficiency in plants[34], so we also tested the separated forms of Cas9 and TREX2 with tandemly expressed Cas9 and TREX2 linked by a self-cleaved T2A peptide (T2A-TREX2). T2A-TREX2 showed increased editing efficiency, as well as decreased junction bias and fewer translocations, at all tested sites; however, T2A-TREX2 did not perform better than Cas9X2 at most of these sites (Fig. 3d-k and Supplementary Figs. 3).

**Cas9TX is a potentially safer Cas9X2 variant for genome editing**. TREX2 plays a role in DNA repair in many cell types, and ectopic overexpression of TREX2 has no impact on cell survival or the cell cycle[37,38]. To further improve the safety of Cas9X2, we generated a Cas9X2 variant with R163A, R165A, and R167A mutations (TREX2-3R) to abolish the DNA-binding activity of TREX2[39]. We then purified TREX2, TREX2-3R, and their *Sp*Cas9-fused forms to perform an in vitro digestion assay with 38-nt oligos (Supplementary Fig. 4a). TREX2-3R showed severely reduced exonuclease activity on oligos in comparison with TREX2 (Supplementary Fig. 4b). Similarly, digested products shorter than 37 nt were detected following exposure to 0.5 nM Cas9X2 for 21 min, but they were not detected following exposure to 2.7 nM Cas9-TREX2-3R (Cas9TX), even after 63 min (Fig. 4a), suggesting that Cas9TX has a better safety profile.

We next applied Cas9TX for genome editing in HEK293T cells and performed PEM-seq analysis. Despite the loss of the DNA-binding capacity of TREX2, Cas9TX was accurately localized to the *HBA1* and *C-MYC2* target sites, at which it had editing

efficiency slightly higher than that of *Sp*Cas9. Similarly, Cas9X2 also showed editing efficiency higher than that of *Sp*Cas9 (Fig. 4b). Both Cas9X2 and Cas9TX efficiently eliminated junction bias and reduced the total translocations in similar levels (Fig. 4c, Supplementary Fig. 4c, d). We next assessed the performance of Cas9TX and Cas9X2 at twelve other sites in the genomes of HEK293T cells. In comparison with *Sp*Cas9, Cas9TX, had significantly higher editing efficiency and induced fewer off-target translocations at the twelve selected sites (4.0–51.2 times), similar to Cas9X2 (Fig. 4d, e, Supplementary Table 6). Moreover, Cas9TX induced a dramatically reduced number of general translocations in comparison with *Sp*Cas9 (Fig. 4f; 2.7–12.3 times).

Furthermore, we also compared chromosomal large deletions induced by Cas9 with those induced by Cas9X2 and Cas9TX. Taking the *C-MYC2* locus as an example, junctions derived from large deletions were distributed widely in the 3 kb region downstream of the target site in Cas9-edited cells. However, in comparison with Cas9, Cas9X2 and Cas9TX reduced large deletions by 7.9- and 9.4-fold in the 3 kb region, and large deletions were mainly confined to the 0.2 kb region downstream of the target site (Supplementary Fig. 4e). For the other tested loci, both Cas9TX and Cas9X2 dramatically reduced the number of large deletions by 3.4- to 55.5-fold in the 3 kb downstream region in comparison with Cas9 (Fig. 4g).

Collectively, these results indicate that Cas9TX can largely suppress chromosomal translocations and large deletions, similar to Cas9X2, but with a lower level of non-specific contact with DNA and a correspondingly improved safety profile.

**Cas9TX induces a slightly higher level of translocations in comparison with base editors**. The cytosine base editor (CBE) and adenine base editor (ABE) systems have been developed to induce point mutations at target sites[40,41]. Since a DSB is not required for CBE or ABE, translocations are rarely formed by these base editing systems. To compare the translocation levels induced by Cas9TX and commonly used base editors, we applied CRISPR-Cas9, CRISPR-Cas9TX, the cytosine base editor BE4max, and the adenine base editor ABEmax[42] to target five sites within the *EMX1*, *DNMT1*, *C-MYC*, *RAG1*, and *BCL11A* genes in HEK293T cells, after which PEM-seq was applied to detect translocations. The BE4max system efficiently targeted multiple cytosines within the editing windows, while the ABEmax system targeted multiple adenines, as exemplified by the *RAG1C* library in Fig. 5a. Of note, the *EMX1* and *C-MYC2* sites were not targetable by ABEmax due to the lack of adenines within the editing windows. In comparison with base editors, *Sp*Cas9-induced substantial indels, with the highest level of base loss neighboring the break site. Interestingly, Cas9TX showed an accumulation of base loss at the broken ends containing the 17 bp truncated sgRNA (Fig. 5a), resulting from end processing by resident Cas9TX after cleavage[29]. According to PEM-seq and CRISPResso, *Sp*Cas9 and Cas9TX generally had higher editing efficiency in comparison with BE4max and ABEmax at the tested sites (Fig. 5b and Supplementary Fig. 5a; ref. [43]).

*Sp*Cas9 induced substantial translocations in all five tested sites, and Cas9TX showed very low translocation levels (Fig. 5c, d, and Supplementary Fig. 5b), consistent with the results described above. In comparison with *Sp*Cas9 and Cas9TX, both BE4max and ABEmax induced extremely low levels of general translocations at the tested sites, but the translocation levels were greater than the background level (Fig. 5b, c, and Supplementary Fig. 5b). Although the translocation level induced by Cas9TX was less than 0.45% at all of the test sites, these levels were still slightly higher than those of the two base editor systems (Fig. 5c and

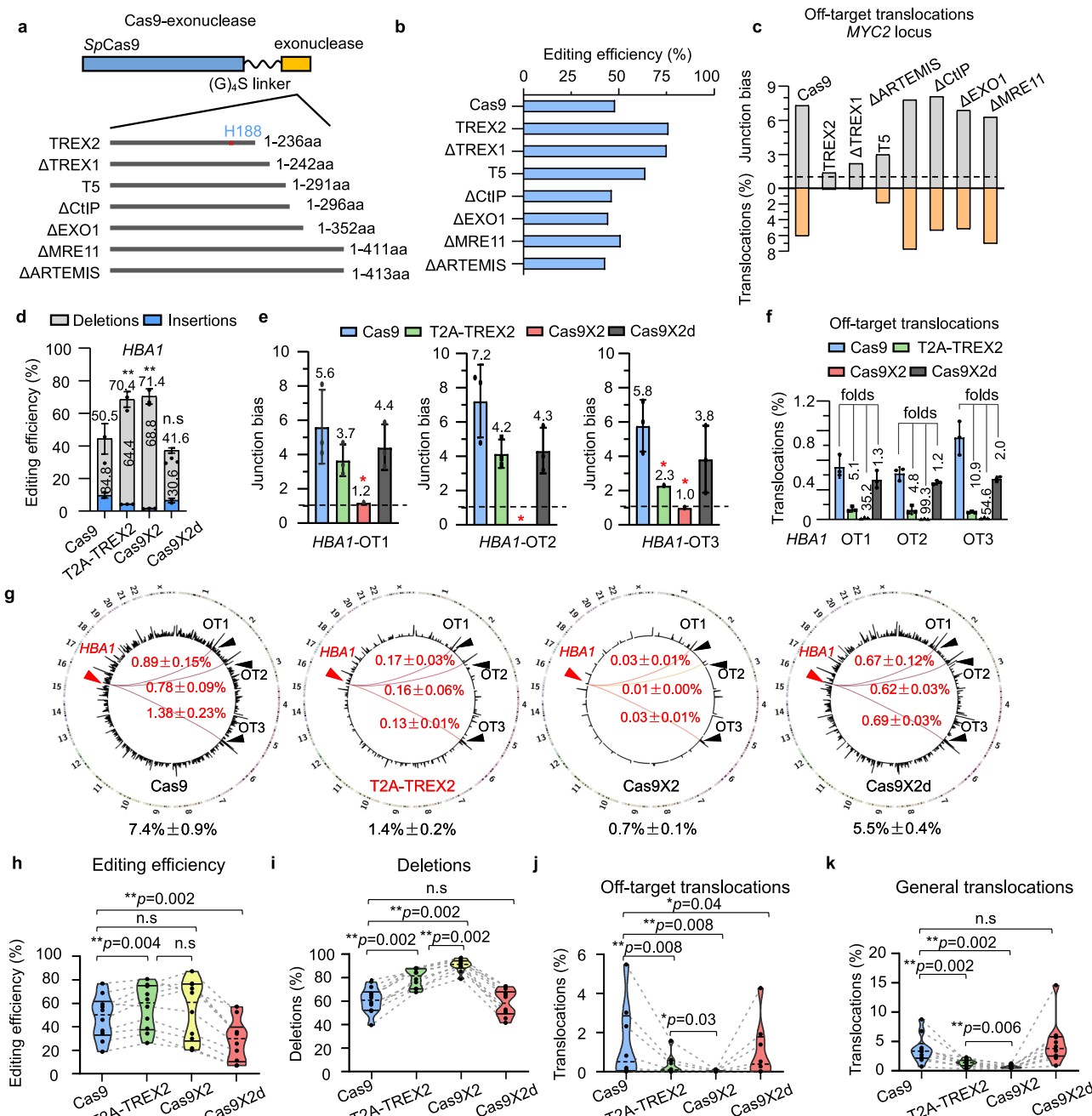

**Fig. 3 Reducing repeated cleavage and chromosomal translocation with a Cas9 exo-endonuclease. a** Schematic for tested Cas9-exonuclease fusion proteins. H188 is indicated. Δ represents truncated. **b, c** Editing efficiency (**b**), junction bias (**c**) at off-target site and percentages for off-target translocations for indicated Cas9-exonucleases in HEK293T cells detected by PEM-seq, N = 1. **d** Editing efficiency for *Sp*Cas9, Cas9X2, T2A-TREX2, and Cas9X2d at the *HBA1* locus in HEK293T cells detected by PEM-seq. Gray bars represent the ratios of deletion, and blue bars represent the ratios of insertion. The deletion ratio and total editing efficiency are marked on each bar. Mean ± SD from three biological replicates, paired two-tailed *t*-test, **p < 0.01, n.s means no significance. **e, f** Junction bias (**e**) and off-target translocation ratios (**f**) for *Sp*Cas9, Cas9X2, T2A-TREX2, and Cas9X2d at *HBA1* in HEK293T cells detected by PEM-seq. The junction bias and fold changes of off-target translocations are at the top of each bar, mean ± SD from three biological replicates. **g** Circos plot for translocations of *Sp*Cas9, Cas9X2, T2A-TREX2, and Cas9X2d PEM-seq libraries at *HBA1* in HEK293T cells. Red arrows indicate on-targets and black arrows indicate identified off-targets. The percentages of off-target translocations are marked in red and the percentages of general translocations are marked in black, mean ± SD from three biological replicates. **h–k** Editing efficiency (**h**), deletions ratios in editing events (**i**), percentages of off-target translocations (**j**), and percentages of general translocations (**k**) for *Sp*Cas9, Cas9X2, T2A-TREX2, and Cas9X2d PEM-seq libraries in HEK293T cells, one PEM-seq library for each locus, N = 10. The loci used were *DNMT1-1*, *DNMT1-2*, *EMX1*, *HBA1*, *C-MYC1*, *C-MYC2*, *C-MYC3*, *RAG1A*, *RAG1B*, and *RAG1C*. Note that off-target translocations were not detected in several loci for Cas9X2. Two-tailed Wilcoxon test, *p < 0.05, **p < 0.01, n.s means no significance. Source data are provided as a Source Data file.

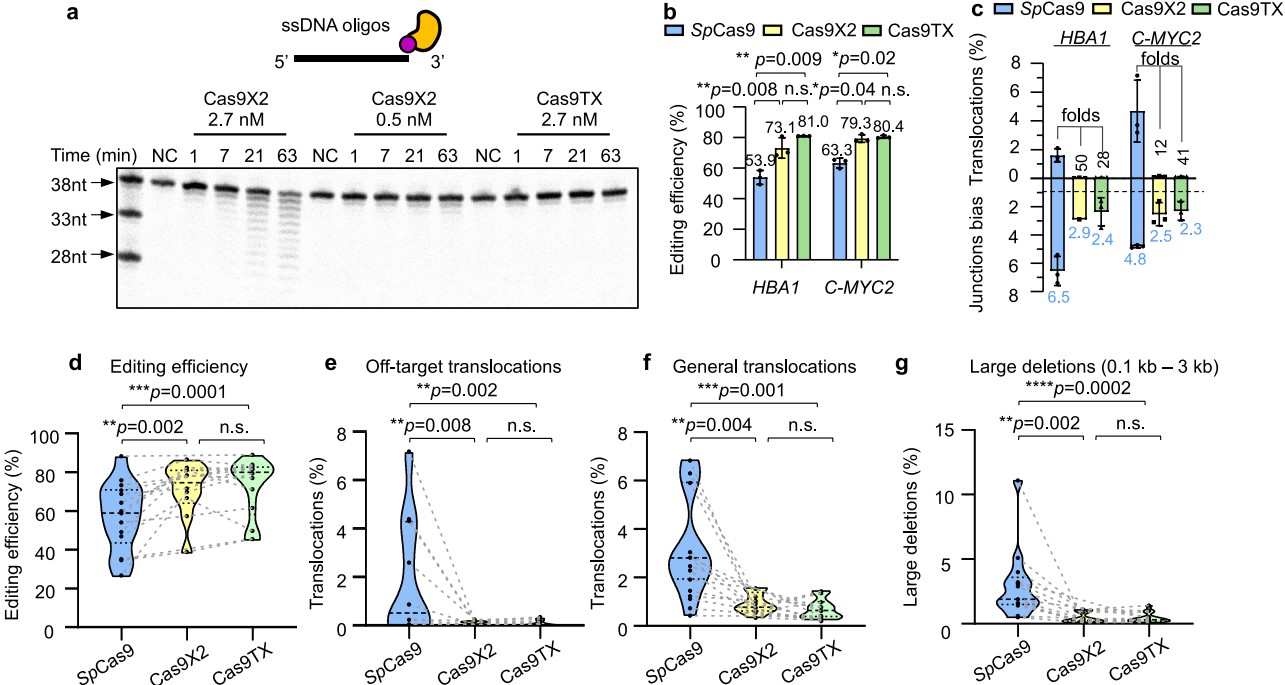

**Fig. 4 Cas9TX is a potentially safer form of Cas9X2. a** Digestion of a 38-mer oligo by Cas9X2 and Cas9TX without sgRNA via in vitro cleavage assay. The indicated amounts of the proteins were incubated with 7.5 nM 38-nt oligos for the indicated time. BSA was the negative control (NC). See methods for more details. **b, c** Editing efficiency (**b**), percentage of off-target translocations (**c**), and junction bias (**c**) for SpCas9, Cas9X2, and Cas9TX at HBA1 and C-MYC2 detected by PEM-seq in HEK293T cells. The HBA1 off-target shown is OT3 as listed in Fig. S4C. The fold changes of off-target translocations and calculated junction bias are at the top of each bar. Mean ± SD from three biological repeats, paired two-tailed t-test, *p < 0.05, **p <0.01, n.s means no significance. **d–g** Editing efficiency (**d**), percentages of off-target translocations (**e**), percentages of general translocations (**f**), and large deletions (**g**) for SpCas9, Cas9X2, and Cas9TX in HEK293T cells detected by PEM-seq, one independent PEM-seq library for each locus, N = 14. The loci used were DNMT1-1, DNMT1-2, HBA1, C-MYC1, C-MYC2, C-MYC3, RAG1A, RAG1B, RAG1C, PTEN, TP53, TRAC, TRBC, and PDCD1. Two-tailed Wilcoxon tests, **p < 0.01 and ***p < 0.001. Source data are provided as a Source Data file.

Supplementary Fig. 5c). Moreover, Cas9TX induced a low level of off-target translocations, which were an extremely rare occurrence with the base editors (Supplementary Fig. 5d). However, considering that the cutting efficiency of the base editors was lower than that of Cas9TX, and multiple targetable bases were counted for the calculation of editing efficiency, the gap between Cas9TX and BE4max was further reduced when they were normalized to the same editing efficiency (Supplementary Fig. 5e). Therefore, Cas9TX may suppress general translocations nearly as well as the base editing system when achieving the same level of gene disruption.

**Cas9TX shows undetectable collateral damage activity**. To further test the safety of Cas9TX for genome editing, we expressed SpCas9-P2A-mCherry or Cas9TX-P2A-mCherry without sgRNA in HEK293T cells. The self-cleaved peptide P2A was included to ensure that the cells expressing SpCas9 or Cas9TX would be marked by mCherry. We then marked DSB signals in the cells using an anti-γH2AX antibody at 24 h post-transfection. The topoisomerase inhibitor etoposide induced substantial DSBs marked by γH2AX, as anticipated (Fig. 6a, b; ref. [44]). The DSB levels were not significantly elevated in SpCas9- and Cas9TX-expressed cells in comparison with the mCherry control (Fig. 6a, b), implying that SpCas9 and Cas9TX did not produce collateral genomic damage activity in the absence of sgRNA. To validate these findings, we performed whole-genome sequencing (WGS) analysis in edited mESC cells. Single cells were sorted into a 96-well plate by flow cytometry 1-day post Cas9:Wrap53 or Cas9TX:Wrap53 transfection. Single

cells were cultured for about 3 weeks and subjected to Sanger sequencing analysis to confirm the editing process (Supplementary Fig. 6a). Three individual clones of the non-edited cells, Cas9-edited cells, and Cas9TX-edited cells were then analyzed by WGS (Supplementary Fig. 6b). The levels of indels were equivalent between Cas9-edited cells and Cas9TX-edited cells (Fig. 6c), further demonstrating Cas9TX has limited collateral damage activity.

We next tested the impact of Cas9TX on other DSBs by co-expressing AsCas12a and Cas9TX with a crRNA for AsCas12a targeting a site within the C-MYC gene (C-MYC3) in HEK293T cells. We performed PEM-seq analysis with a bait primer at the AsCas12a target site and found that the editing efficiency of AsCas12a was relatively higher when it was co-expressed with Cas9TX in comparison with SpCas9 (18.4% vs 15.7%; Fig. 6d). Moreover, fewer total translocations for AsCas12a:C-MYC3 were observed when AsCas12a was co-expressed with Cas9TX in comparison with SpCas9 (1.4% vs 2.6%; Fig. 6d), suggesting that Cas9TX can also enhance editing efficiency and eliminate translocations for other co-expressed editing enzymes.

We next employed PEM-seq with a bait primer at the identified SpCas9:C-MYC2 off-target site to examine the impact of Cas9TX on cleavage of off-target sites. We detected a higher editing frequency at the off-target site with Cas9TX in comparison with SpCas9. However, the editing efficiency at the off-target site was proportional to the editing efficiency at the MYC2 on-target site for both SpCas9 and Cas9TX (Fig. 6e). Moreover, the MYC2 off-target showed a level of translocations with SpCas9 14.6-times as that with Cas9TX (Fig. 6f). We also employed tracking of

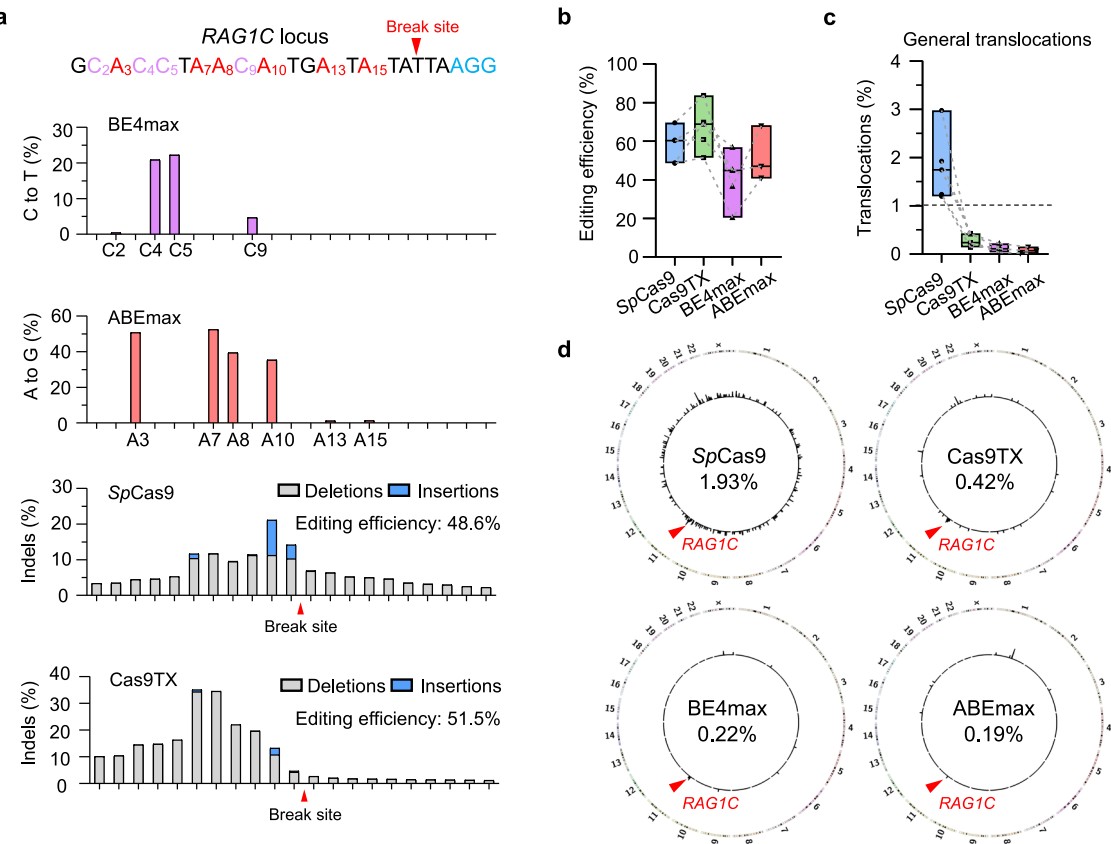

**Fig. 5 Suppressing chromosomal translocations down to the level of base editors by Cas9TX. a** Editing patterns around the break site for BE4max, ABEmax, *Sp*Cas9, and Cas9TX at the *RAG1C* locus detected by PEM-seq. Red arrows indicate the presumed break sites. Accumulative levels of mutations, deletions, and insertions are shown at nucleotide resolution. **b**, **c** Editing efficiency (**b**) and the percentages of general translocations (**c**) for *Sp*Cas9, Cas9TX, BE4max, and ABEmax detected by PEM-seq at *EMX1*, *C-MYC2*, *DNMT1-2*, *RAG1C*, and *BCL11A* in HEK293T cells. One independent PEM-seq library for each locus, $N = 5$. Of note, the *EMX1* and *C-MYC2* sites were not targetable by ABEmax. Values from minimum to maximum are shown in the box. The line represents the median. **d** Circos plot showing the distribution of translocations for *Sp*Cas9, Cas9TX, BE4max, and ABEmax at *RAG1C* in HEK293T cells. Red arrows indicate the *RAG1C* target site. The percentages of general translocations are shown in the center.

indels by decomposition (TIDE) to roughly measure the editing frequency at multiple off-target sites in HEK293T cells in which the *VEGFA* or *EMX1* sites were targeted by *Sp*Cas9 or Cas9TX. The vast majority of the tested off-targets showed increased editing frequencies with Cas9TX, but the frequencies were also proportional to the editing efficiency of the on-target sites (Fig. 6g). These data indicate that Cas9TX can not only enhance cleavage at both on- and off-target sites at similar levels but can also effectively prevent the translocations for off-targets or other genome editing enzymes.

**Cas9TX eliminates translocations in engineered CAR T cells**. To test the ability of Cas9TX to eliminate translocations in engineered CAR T cells, we transduced the CD19-4-1BB-CAR-encoding retrovirus[45] into activated T cells and then applied *Sp*Cas9 or Cas9TX to edit the *TRAC*, *TRBC*, or *PDCD1* genes via RNP delivery (Fig. 7a). The transduction efficiency of the CAR retrovirus was very similar for the *Sp*Cas9 and Ca9TX populations (Fig. 7b). The editing efficiency of CRISPR-Cas9 and CRISPR-Cas9TX was 40–60%, and about half of the T cells lost TCR 3-days post-transfection (Fig. 7c, d, Supplementary Table 7). The levels of chromosomal translocations among *TRAC*, *TRBC*, and *PDCD1* target sites were greatly reduced in CRISPR-Cas9TX-edited T cells in comparison with CRISPR-*Sp*Cas9-edited cells (Fig. 7e, f). The percentage of *TRAC-PDCD1* translocations was decreased from 0.23 to 0.01%, and the percentage of *TRAC-TRBC*

translocations was decreased from 0.17 to 0.02% (Fig. 7g, h). Other translocations among the three on-target genes also showed huge decreases (10–30 times) in Cas9TX-edited T cells at 3-days post-transfection (Supplementary Figs. 7a-d). The translocations rates were further declined at 7- or 14-days post-transfection and finally arrived at one out of 2000–50,000 cells (Fig. 7g, h, and Supplementary Fig. 7a-d). Moreover, no translocation junction was observed for the *TRAC* off-target site in the Cas9TX-edited T cells (Fig. 7f). The rate of general translocations in Cas9TX-edited T cells was reduced significantly at 3-days post-transfection in comparison with that of CRISPR-Cas9-edited T cells, and this rate was further reduced at 7- and 14-days post-transfection (Fig. 7i, Supplementary Fig. 7e, f).

To test the ability of *Sp*Cas9-edited or Cas9TX-edited CAR T cells to kill CD19+ K562 cells, we mixed engineered CAR T cells with CD19+ K562 cells at different ratios. Equal numbers of CD19− K562 cells with CD19+ cells were also included in the system for final normalization[45]. Cas9TX-edited T cells showed CAR ability similar to that of the *Sp*Cas9-edited cells. Approximately 72% of CD19+ K562 cells were lysed when they were mixed with CAR T cells at a 1:1 ratio, and nearly 100% of CD19+ K562 cells were lysed when they were mixed at a ratio of 1:5 (Fig. 7j and Supplementary Fig. 7g). These results show that CRISPR-Cas9TX can eliminate chromosomal translocations in CAR T cells without affecting their CD19 targeting function.

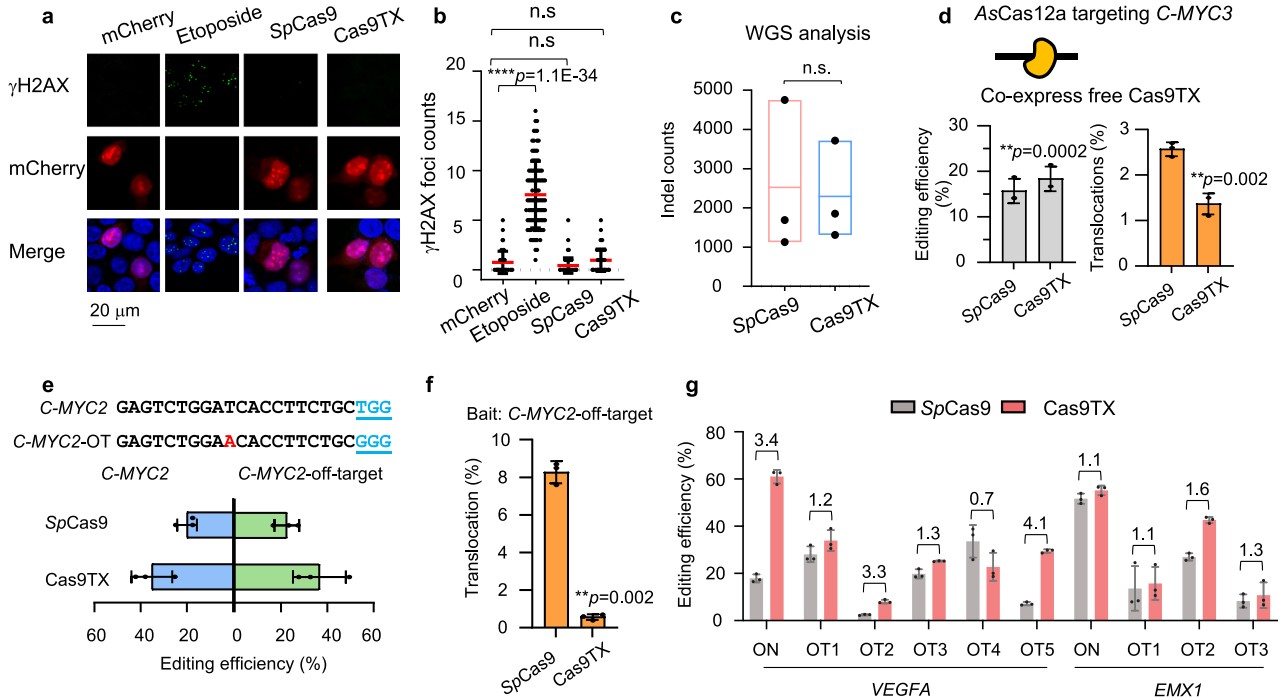

**Fig. 6 Effect of Cas9TX on genome stability. a** Immunofluorescence for γH2AX in HEK293T cells treated with etoposide, SpCas9, or Cas9TX. Images were taken 24 h after transfection by confocal microscopy. Both SpCas9 and Cas9TX were expressed with a P2A-mCherry tag. mCherry alone was the negative control. **b** Statistics for γH2AX foci in each HEK293T cell treated with etoposide, SpCas9, or Cas9TX. N = 100 cells were analyzed for each group, two-tailed t-test, ****p < 0.00001; n.s, no significance. **c** Indel counts for Cas9-edited and Cas9TX-edited mESCs by WGS analysis. Three biological replicates, two-tailed t-test. n.s means no significance. Values from minimum to maximum are shown in the box. The line represents the average. **d** The impacts of SpCas9 and Cas9TX on the editing efficiency and translocation levels of AsCas12a:C-MYC3 detected by PEM-seq in HEK293T cells. Mean ± SD from three biological replicates. Two-tailed t-test, **p < 0.01. **e** Editing efficiency for SpCas9 and Cas9TX at C-MYC2 and the identified C-MYC2 off-target site detected by PEM-seq in HEK293T cells. Mean ± SD from three biological replicates. Two-tailed t-test, **p < 0.01. DNA sequences for C-MYC2 and the C-MYC2 off-target are shown at the top. Mismatched DNA is in red. **f** Percentages of translocations cloned from the C-MYC2 off-target site by PEM-seq in HEK293T cells. Mean ± SD from three biological replicates. Two-tailed t-test, **p < 0.01. **g** Editing frequency of SpCas9 and Cas9TX at the on-targets and off-targets of VEGFA and EMX1 identified by PEM-seq analysis in Figs. 3 and 4. Primers were designed at off-target sites with PCR amplification. Editing efficiency was evaluated by TIDE. Fold changes for Cas9TX to SpCas9 are at the top. Mean ± SD from three replicates. Source data are provided as a Source Data file.

## Discussion

DSBs induced by CRISPR-Cas9 are subjected to different DNA repair pathways to generate a spectrum of DNA repair outcomes. The non-homologous end-joining (NHEJ) pathway directly joins two intact or lightly processed broken ends together to generate perfectly rejoined products or small indels[46]. It has been reported that more than 50% of repair outcomes following CRISPR-Cas9 cleavage are perfect rejoinings based on sequencing data at two neighbor target sites[29,47]. The perfect rejoinings are indistinguishable from the target sequences and can be repeatedly cleaved by CRISPR-Cas9. During each round of cleavage, a few DSBs can escape the surveillance of the DNA damage response and lead to chromosomal translocations. Therefore, the number of translocations increases gradually throughout the gene editing process, leading to significant accumulation (Supplementary Fig. 8a). Of note, while chromosomal translocations can occur between SpCas9-induced DSBs, they can also occur among general DSBs arising during various cellular activities[12,19,28,48].

Chromosomal translocations have been widely observed in many types of CRISPR-Cas9-edited cells[10,15,16,21,49]. Based on our PEM-seq analysis, we determined that translocations occurred between two target genes in CRISPR-Cas9-targeted T cells at a frequency of one in 50–300 edited T cells (Fig. 1c). However, the translocation frequency between two I-SceI target sites was reported to be lower, at approximately one in 300–1200 cells[18], probably because I-SceI-induced sticky DSB ends are more

inclined to be processed. Similarly, CRISPR-Cas12a (or AsCpf1) with 4 bp sticky ends has been found to induce fewer translocations in comparison with CRISPR-Cas9[50]. Regarding chromosomal translocations, gene editing at strong enhancers or oncogenes might be risky. In this context, translocations involving strong enhancers from antigen receptor loci or c-Myc have been extensively investigated as drivers of tumorigenesis in developing lymphocytes[12,28,51]. In addition to the translocations observed in TRAC and TRBC in engineered T cells from this study, translocations arising in hematopoietic stem and progenitor cells (HSPCs) during CRISPR-Cas9-mediated editing of the CCR5 and B2M genes[10,15,16,21] can also pose a threat to the genome integrity of stem cells and affect both the circulatory and immune systems.

The fusion of an exonuclease with SpCas9 allows immediate end processing at cleavage and can consequently reduce the percentage of intact broken ends (Fig. 3). Therefore, a large portion of perfect rejoinings are transformed into indels at the initial cleavage. In this context, Cas9TX can slightly increase editing efficiency and maintain the number of translocations at the background level (Supplementary Fig. 8a, b) in single-site (Fig. 4) and multiplex genome editing scenarios (Fig. 7). We showed that Cas9TX is compatible with RNP, and the small size of TREX2-3R (236 amino acids) should allow Cas9TX to be packaged in split adeno-associated viruses (AAV; ref. [52]).

We found that CRISPR-Cas9TX nearly eliminated chromosomal translocations between TRAC, TRBC, and PDCD1 target sites

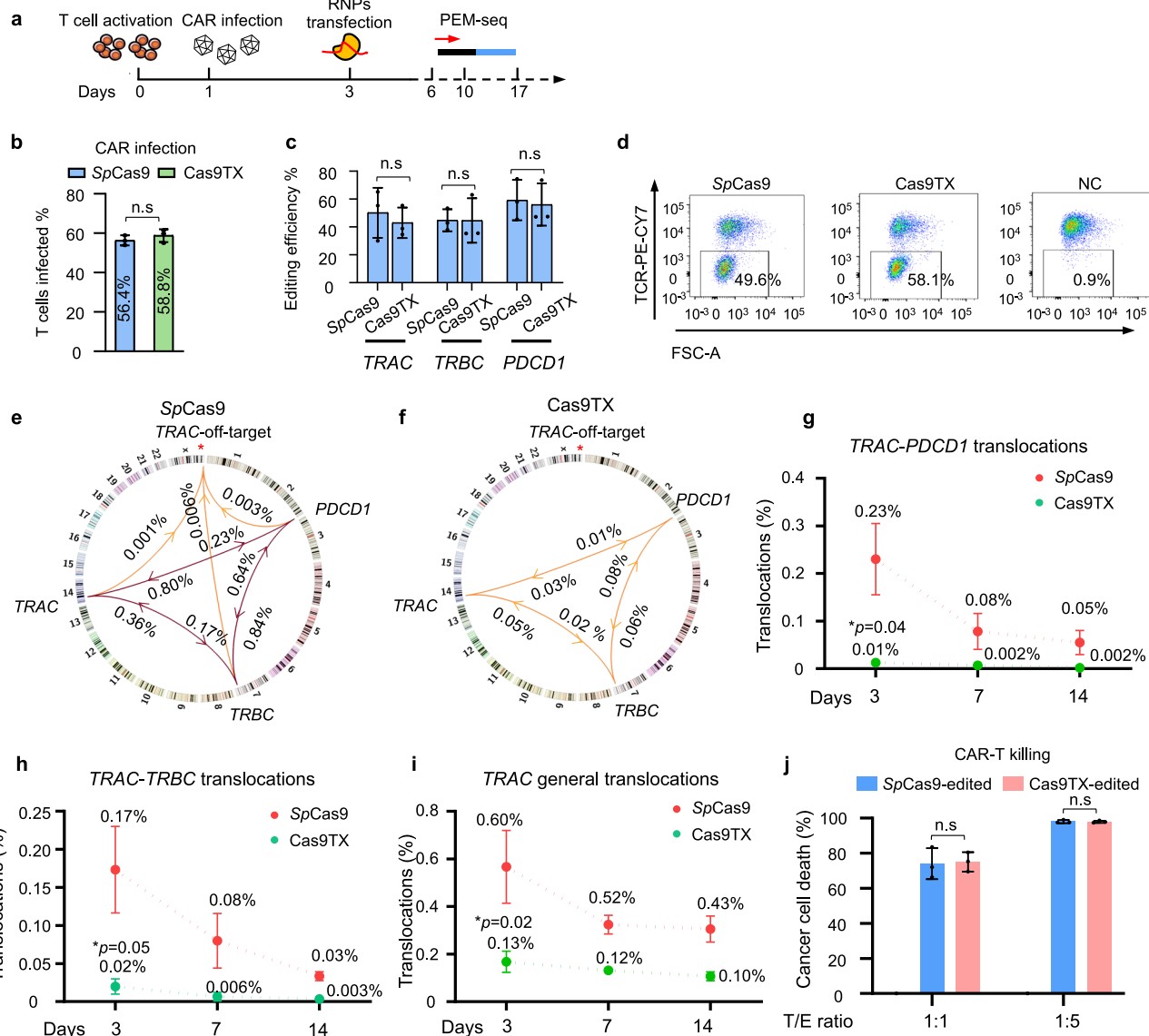

**Fig. 7 Elimination of chromosomal translocations by Cas9TX in CAR T cells. a** Schematics for the manufacturing of CAR-T cells by *Sp*Cas9 or Cas9TX RNPs. Human T cells obtained from human cord blood were activated for 3 days by anti-CD3/CD28, infected by a lentivirus carrying anti-CD19 scFv, and transfected with *Sp*Cas9/Cas9TX RNPs. CAR-T killing assays were performed at 3-days post-transfection, and translocations among *TRAC*, *TRBC*, and *PDCD1* were analyzed by PEM-seq at 3-, 7-, and 14-days post-transfection. See methods for more details. **b** Percentages of T cells infected by the lentivirus carrying CAR-CFP detected by FACS. Paired two-tailed *t*-test, mean ± SD from three biological replicates; n.s, no significance. **c** Editing efficiency at the *TRAC*, *TRBC*, and *PDCD1* genes at 3-days post-transfection detected by PEM-seq. Paired two-tailed *t*-test, mean ± SD from three biological replicates; n.s, no significance. **d** Representative flow cytometry results indicating TCR expression levels at 3-days post-transfection. NC is the negative control without RNP editing. **e**, **f** Circos plots indicating translocations among *TRAC*, *TRBC*, *PDCD1*, and *TRAC* off-targets by *Sp*Cas9 (**e**) or Cas9TX (**f**) at 3 days after RNP transfection detected by PEM-seq. Mean values for three biological replicates are indicated. Red asterisks indicate the identified *TRAC* off-target site, depicted as described in the legend to Fig. 1c. **g–i** The ratios of *Sp*Cas9- or Cas9TX-induced translocations of *TRAC-PDCD1* (**g**), *TRAC-TRBC* (**h**), and *TRAC* general translocations (**i**) cloned from *TRAC* by PEM-seq at 3-, 7-, and 14-days post-transfection. Mean values for three biological replicates are indicated. Mean ± SD, paired two-tailed *t*-test, *p* < 0.05, **p* < 0.01. **j** The killing capacities of *Sp*Cas9- and Cas9TX-editing CAR-T cells were evaluated by CD19[+] K562 killing assay. T indicates target cells and E indicates effector cells. Mean ± SD from three biological replicates, two-tailed *t*-test, n.s means no significance.

while having an undetectable impact on the killing ability of CAR, outperforming CRISPR-Cas9 (Fig. 7). In comparison with Cas9TX, high-fidelity *Sp*Cas9 variants are not able to suppress translocations. With regard to base editors that are frequently used for gene corrections, but are rarely applied for gene disruption, both BE4max and ABEmax induce few translocations because nicks are converted to DSBs at a low frequency[19]. The level of general translocations induced by CRISPR-Cas9TX is only slightly higher than that of BE4max or ABEmax (Fig. 5c). Of

note, base editors can also target RNA[53,54], but the CRISPR-Cas9TX system does not possess this capability. In conclusion, the application of CRISPR-Cas9TX for genome editing is expected to ensure relatively good genome integrity comparable to that achieved with commonly used base editor systems.

## Methods

**T cell culture and electroporation**. Human primary T cells were obtained from human cord blood using negative selection kits (STEMCELL Technologies, cat.

no.19051). Cord blood cells were obtained from the Cord Blood Bank of Beijing. T cells were cultured in RPMI 1640 (Corning) with 30 units/mL recombinant human interleukin-2 (IL-2, Gibco, cat. no. PHC0021), glutamine (Corning), 15% fetal bovine serum (FBS, Excell Bio), and penicillin/streptomycin (Corning) at 37 °C under 5% $CO_2$. Before Cas9 RNP transfection, T cells were activated with a 1:1 ratio of human anti-CD3/28 beads for 3 days (Thermo Fisher, cat. no.11161D) (20 µl in 1 ml culture medium for 1 million T cells). For electroporation, 9 µg SpCas9/Cas9TX and 6 µg sgRNA (2 µg for TRAC, TRBC, and PDCD1 respectively, sgRNAs were synthesized with 2′-O-methyl and phosphorothioate modifications in GeneScript) were mixed for 15 min at RT (room temperature) and then transfected into 1 million T cells using a Celetrix Electroporator with parameter sets as follows: $V_{set}$ = 420 V, $T_{set}$ = 20 ms, $P_{num}$ = 1 N, $T_{int}$ = 1 ms. T cells were recovered in the culture without bead stimulation for 1 day, after which anti-CD3/28 beads were added for further experiments. PE/Cyanine7 anti-human TCR α/β (BioLegend, cat. no. 306719) was used to check the TCR knockout efficiency. FACS data were analyzed by FlowJo (version X 10.0.7).

**Lentivirus package and human T cell infection**. First, 5 µg pMD2.G (Addgene 12259), 10 µg psPAX2 (Addgene 12260), and 20 µg pCD19 scFV 4-1BB plasmids were co-transfected into cultured HEK293T cells in a 10 cm dish using PEI (Sigma, P3143). Supernatants were filtered by a 0.45 µm filter and were then concentrated at 70,000 × g for 2 h at 4 °C using a Beckman Optima L8-80XP centrifuge at 60 h post-transfection. Activated human T cells (1-day post-stimulation) were infected by lentivirus at 500 × g for 2 h with 8 µg/mL polybrene (Sigma, TR-1003).

**Purification of SpCas9, SpCas9 high-fidelity variants, TREX, TREX-3R, Cas9X2, and Cas9TX**. SpCas9, SpCas9 high-fidelity variants, Cas9X2, and Cas9TX were expressed using the pET28a backbone (Addgene 53135). The TREX2 and TREX2-3R mutants were expressed using pDB-His-MBP (Addgene 123365).

For the expression of the MBP-TREX2 fusion proteins, the pDB-His-MBP-TREX2 plasmid was transfected into E. coli BL21 (DE3) Rosetta cells. The procedures were based on a previous method[37], except that MBP cleavage was performed using TEV instead of Genenase.

For the expression of SpCas9, SpCas9 variants, Cas9X2, and Cas9TX, the pET28a plasmid was transformed into E. coli BL21 (DE3) Rosetta cells and expressed by IPTG (Amresco, 0487) induction. The cells were lysed by sonication in lysis buffer (20 mM HEPES, pH 7.5, 10% glycerol, 0.1% Triton X-100) containing 1 mM PMSF, and cell debris was removed by centrifugation at 20,000 × g for 1 h. The supernatant was loaded onto a HisTrap HP column (GE Healthcare) and eluted with an imidazole gradient of 0–300 mM in lysis buffer. Next, the components were subjected to gel filtration in a Superdex 200 column (GE Healthcare). Finally, the purified proteins were quantified using a BSA protein standard and stored at −80 °C in lysis buffer before use.

**PCR for TRAC-TRBC, TRAC-PDCD1, TRBC-PDCD1 translocations**. For first-round PCR, primers GTGTCACAAAGTAAGGATTCTG and CTAGTCTTGTCTGC TACCTGGATC were used for the TRAC-TRBC translocation amplification, primers GTGTCACAAAGT AAGGATTCTG and GCACCCTCCCTTCAACCTGACCTGGGAC were used for the TRAC-PDCD1 translocation amplification, and primers CTAGTCTTGTCTGCTACCTGGATC and GCACCCTCCCTTCAACCTGACCTGGGAC were used for the TRBC-PDCD1 translocation amplification. PCR products were recycled and underwent second-round PCR with primers TTCTGATGTGTATA TCACAG and CTAGTCTTGTCTGCTACCTGGATC for the TRAC-TRBC translocation amplification, primers TTCTGATGTGTATATCACAG and GAGAAGGCGGCAC TCTGGTG for the TRAC-PDCD1 translocation amplification, and primers CTAGTCTTGTCTGCTACCTGGATC and GCTCACCTCCGCCTGAGCAG for the TRBC-PDCD1 translocation amplification. Original scans are provided in the Source Data File.

**PEM-seq analysis**. The PEM-seq libraries were generated as described previously[9,55]. In general, biotinylated primer was set within 150 bp from Cas9 target site to achieve primer extension. Biotinylated products were enriched by Dynabeads™ MyOne™ Streptavidin C1 (Thermo Fisher, 65001) followed by bridge adapter ligation. Then locus specific primer was used for nested PCR followed by Illumina Hiseq sequencing. To better analyze translocation with the PEM-seq pipeline, we developed a new translocation filter module to filter the false translocation junctions, including those with the same random molecular barcodes (RMBs) and the same junction sequence with highly similar RMBs (<2 mismatches)[22]. Briefly, PEM-seq can identify genome editing products: perfect rejoinings, indels, translocations, and other chromosomal abnormalities. The ratio of indels to total identified products was defined as the editing efficiency. Indels were defined as deletions (<100 bp) and insertions (<20 bp). For base editors, editing efficiency was calculated by counting all products identified by CRISPResso (version 1.0.8) (>0.2%) using default parameters. The editing frequency at the main cytosine or adenine was used as the "desired" editing efficiency of BE4max or ABEmax and normalized to the same editing efficiency with SpCas9 and Cas9TX, as shown in Supplementary Fig. 5e.

Translocation hotspots with a sequence highly similar to the target site (≤8 nt mismatches considering both sgRNA and PAM sequences) and with junctions at the presumable SpCas9 cut-site were considered as off-target sites. General translocations were calculated by excluding junctions ±20 kb around the target sites and ±100 bp around the off-target sites.

For the calculation of junction bias in human T cells, HEK293T, and K562 cells, junctions located within ±100 bp of an off-target site were counted for the PAM-distal and PAM-proximal ends. Note that for the statistics for junction bias in Supplementary Fig. 2c, the junctions of all RAG1A off-targets were combined to calculate the bias because of the low number of translocation junctions. The primers used for PEM-seq are listed in Supplementary Table 8.

**In vitro digestion of DNA fragments by SpCas9**. In general, sgRNA fused with scaffold RNA was transcribed by using a T7 High-Efficiency Transcription Kit (TransGen Biotech) in vitro. A 100 nM concentration of SpCas9 and 300 nM RNA were included in each reaction. DNA fragments were digested under the following conditions: 20 mM HEPES (pH 7.5), 5% glycerol, 100 mM KCl, 1 mM dithio-threitol, 10 mM $MgCl_2$, and 0.5 mM EDTA at 37 °C for 2 h.

For the DNA substrates, the SpCas9:TRAC on-target site and its off-target site were amplified using the primers shown in Supplementary Table 8. Constructs consisting of the TRAC on-target site joined with two ends of the TRAC off-target (retargetable and untargetable translocation products) were generated via overlap PCR.

**Arrest of K562 cells in G1 phase**. K562 cells were treated with 5 µM palbociclib (PD-0332991; Selleck, S1116) for 36 h before SpCas9 transfection and were further cultured in 5 µM palbociclib before cell harvest. For cell cycle analysis, cells were labeled with 50 µM BrdU for 60 min and fixed by paraformaldehyde (PFA) for 60 min at 4 °C followed by anti-BrdU (100×, BD Biosciences, 556028) incubation for 40 min. Next, cells were stained with 7-AAD (250×, BD Pharmingen, 559925) for 20 min and analyzed by FACS.

**Plasmid construction**. The DNA for TREX2, TREX1, CtIP, EXO1, MRE11, and ARTEMIS was obtained from reverse transcription from human cDNA. The DNA for T5 was synthesized by Genwiz (Beijing, China). And the indicated Cas9-exo-endonuclease plasmids were obtained by Gibson assembly into the pX330 backbone with separately CMV-driven mCherry confirmed by Sanger sequencing analysis. All of the sgRNAs used in this study are listed in Supplementary Table 9. SpCas9, Cas9X2, Cas9X2d, Cas9TX, BE4max (a gift from Dr. Chengqi Yi), and ABEmax (a gift from Dr. Chengqi Yi) were constructed into the pX330 backbone (Addgene 42230) with P2A-mCherry or CMV-driven mCherry. The sgRNA was cloned into a separate pX330 backbone with CMV-driven GFP instead of SpCas9. The TREX2-H188A and TREX2-3R mutants were obtained by overlap PCR and confirmed by Sanger sequencing analysis.

**Cell line culture and transfection**. HEK293T cells (from Dr. Frederick Alt Lab, Harvard Medical School) were cultured in Dulbecco's modified Eagle's medium (Corning) with glutamine (Corning), 10% FBS, and penicillin/streptomycin (Corning) at 37 °C under 5% $CO_2$. K562 cells were cultured in RPMI 1640 (Corning) with glutamine (Corning), 15% FBS, and penicillin/streptomycin (Corning) at 37 °C under 5% $CO_2$. mESCs were cultured in ES-DMEM (Millipore) containing 15% FBS, nonessential amino acids (NEAAs) (Corning), nucleotides (Millipore), penicillin/streptomycin (Corning), glutamine (Corning), PD0325901 (Selleck), CHIR99021 (Selleck), and LIF (Millipore). HEK293T cells were co-transfected with 3 µg of the SpCas9/Cas9X2/Cas9TX/Base editors-expressed plasmid and 3 µg of the sgRNA plasmid by PEI (Sigma) in 6 cm dishes (Figs. 2e, 3, 4, 5, 6g). The SpCas9 plasmid (1 µg per million cells) and sgRNA plasmid (1 µg per million cells) were co-introduced into K562 cells (3111C0001CCC000039, National Infrastructure of Cell line resource, China) using the 4D-Nucleofector System with the FF120 program in SF buffer (Lonza). The same amounts of the Cas9 plasmid and GFP were co-introduced into mESCs (from Dr. Xiong Ji Lab, Peking University) using the 4D-Nucleofector System and the GC104 program in Cytomix buffer. All sample cells were collected via FACS sorting with mCherry and/or GFP. Cell lines have been confirmed by STR (Short tandem repeat).

For repeated cleavage at the TP53 locus in HK293T cells, 3.6 µg plasmids expressing Cas9:TP53-sg1&2 and 3.6 µg plasmids expressing Cas9:C-MYC1 or Cas9:C-MYC2 were co-transfected into HEK293T cells in a 6 cm dish. Genomic DNA was collected at 72 h post-transfection followed by PEM-seq and data analysis. The Tp53-sg2 sequence is GACCATTACTCAGCTCTGAG. The Tp53-sg2 sequence is ACCATTACTCAGCTCTTGAG.

To determine changes in editing efficiency, junctions bias and off-target translocations, 2 µg plasmids expressing Cas9:C-MYC2 were transfected into 1 million K562 cells or 7.2 µg plasmids expressing Cas9:C-MYC2 were transfected into HEK293T cells in 6 cm dishes. Genomic DNA was collected every 6 h post-transfection until 48 h post-transfection, followed by PEM-seq and data analysis.

For Cas9/Cas9TX co-expressed with AsCas12a, 3.6 µg AsCas12a-expressed plasmids targeting C-MYC3 locus and 3.6 µg Cas9/Cas9TX-expressed plasmids were co-transfected into HEK293T cells in a 6 cm dish. Genomic DNA was collected at 72 h post-transfection followed by PEM-seq and data analysis.

**Exonuclease assay-time course reaction**. The reaction assays (10 μL) contained 20 mM Tris-HCl (pH 7.5), 5 mM $MgCl_2$, 2 mM DTT, 100 μg/mL BSA, 7.5 nM 38-mer oligonucleotides (Genewiz), and the TREX2 protein (or Cas9X2). Incubation was performed at room temperature for the indicated period of time. The reactions were stopped by the addition of 30 μL of ethanol and dried *in vacuo*. Pellets were resuspended in 6 μL of 1× loading buffer, denatured at 95 °C for 5 min, and separated in 15% denaturing polyacrylamide gels.

**Immunofluorescence for γH2AX detection**. HEK293T cells were cultured on glass slides in a six-well dish. After *Sp*Cas9/Cas9TX transfection or 10 μM etoposide (Sigma, S1225) treatment for 24 h, the cells on the slides were fixed in 4% PFA for 10 min at RT, followed by washing in PBS and permeabilization with 0.5% TritonX-100 for 15 min. Before primary antibody staining, cells were blocked using 3% FBS for 60 min. The cells were incubated with rabbit anti-γH2A.X (phospho S139) (Abcam, ab2893) at 1:500 dilution for 1 h at room temperature or overnight at 4 °C, followed by washing in 0.2% Tween. Next, the cells were stained with Alexa 488 Fluor goat anti-rabbit IgG HRP (Abcam, ab6721) secondary antibody at a 1:500 dilution for 60 min at RT, and nuclear staining was performed using Hoechst 33342 (Sigma, B2261) at 1 mg/mL for 15 min at RT. Finally, the slides were mounted to microscope slides for microscopy analysis. Images were acquired using a Nikon A1R high-speed laser confocal microscope and analyzed by ImageJ (version 1.51 J8) according to the instructions included with the software.

**Whole-genome sequencing for mESCs**. Two micrograms of Cas9/Cas9TX-expressed plasmid were introduced into 1 million mESCs using the 4D-nucleofector and the GC104 program in Cytomix buffer. Single cells were sorted into a 96-well plate by flow cytometry 1 day after Cas9:*Wrap53* or Cas9TX:*Wrap53* transfection. Cells were cultured for approximately 3 weeks and subjected to Sanger sequencing analysis to confirm gene editing. Genomic DNA was sonicated to form fragments of about 300–500 bp, followed by WGS library construction using the NEBNext® Ultra™ II DNA Library Prep Kit (E7645S) followed by Strelk2 (version 2.8.2) pipeline analysis[56].

**CAR-T killing assay**. K562 cells with CD19 and BFP expression or K562 cells with FITC (without CD19) were co-cultured with *Sp*Cas9/Cas9TX-edited CAR-T cells at an E:T ratio of 1:1, 5:1, or 10:1 for 24 h. The killing efficiency was detected by FACS and analyzed by FlowJo (version X 10.0.7).

**Statistics and reproducibility**. Statistical analysis was performed using two-tailed Student's *t*-test and two-tailed Wilcoxon signed-rank test. Data are presented as the mean ± SD for biological repeats, and $p < 0.05$ was considered significant. For results shown by gels in Fig. 2c, Supplementary Figs. 1c, 4a, b, at least three times was repeated for each experiment, and one was shown in the figure. No statistical method was used to predetermine sample size. The experiments were not randomized. No data were excluded from the analyses and the investigators were not blinded to allocation during experiments and outcome assessment.

**Reporting summary**. Further information on research design is available in the Nature Research Reporting Summary linked to this article.

## Data availability

Original PEM-seq data for off-target bias detection in HEK293T cells were deposited into the NCBI Gene Expression Omnibus (GEO) (GSE116231) and original PEM-seq data for comparison for Cas9 and Cas9TX were deposited into the National Omics Data Encyclopedia (NODE) (OEP000911). Source data are provided with this paper.

## Code availability

Code for PEM-seq analysis is available at the GitHub site: https://github.com/liumz93/PEM-Q.

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

## Acknowledgements

We thank Dr. Chengqi Yi for the base editor plasmids. We thank the Flow Cytometry Core at the National Center for Protein Sciences at Peking University, particularly Liying Du and Yinghua Guo, for technical help. We thank the National Key R&D Program of China (2017YFA0506700), the NSFC (grant 32122018 and 31771485), the SLS-Qidong Innovation Fund, the Clinical Medicine Plus X–Young Scholars Project (No. PKU2020LCXQ021), and the PKU-TSU Center for Life Sciences for funding support. J.H. is a Bayer investigator.

## Author contributions

J.Y., R.L., C.X., and J.H. designed the experiments; J.Y., R.L., C.X., Y.W., W.Z., and W.X. performed the PEM-seq experiments; J.Y. performed the T-cell-associated experiments. X.L., D.L., W.S., P.W., B.X., H.L., and T.L. helped with the T-cell-associated experiments; L.K. helped with the Cas9-timing experiments; J.Y., C.X., R.L., M.L., and J.H. analyzed the data; J.Y., C.X., R.L., and J.H. wrote the paper.

## Competing interests

The authors declare no competing interests.
