## [Peer Review File · Nature Communications]

Reviewers' Comments:

Reviewer #1:

Remarks to the Author:

The authors have successfully addressed my comments and have provided new experiments supporting their conclusions. I therefore recommend publication.

Reviewer #2:

Remarks to the Author:

The authors added new information in the revision. some previous comments were addressed, however there are still remaining concerns:

1. Conceptual: because this is not new, authors need to compare their methods rigorously to previous methods, in a quantitative setting
2. technical: many experiments were done in a single replicate. this makes it impossible to know the distribution, and thereby creating reproducibility issues.

A point-by-point response to the referees' comments (response in blue)

Reviewer #1 (Remarks to the Author):

The authors have successfully addressed my comments and have provided new experiments supporting their conclusions. I therefore recommend publication.

Thanks very much for the referee's time to review our work.

Reviewer #2 (Remarks to the Author):

The authors added new information in the revision. some previous comments were addressed, however there are still remaining concerns:

1. Conceptual: because this is not new, authors need to compare their methods rigorously to previous methods, in a quantitative setting

Thanks for the comment. We agree with the reviewer that rigorous controls are required to draw solid conclusions. In this context, we have confirmed the accuracy for PEM-seq to quantify editing products by comparing it with other methods including RFLP, TIDE, and CRISPResso in our previous studies (Yin et al., 2019, PMID: 30937179; Liu et al., 2021, PMID: 34365511). Secondly, we compared Cas9TX with various Cas9 variants in this study, including Cas9 nuclease (**Figs. 4 and 7**), high-fidelity Cas9 variants (**Fig. 1g**), Cas9-TREX2 (**Figs. 3 and 4**), base editors (**Fig. 5**), Cas9-T2A-TREX2 (**Fig. 3**), as well as Cas9 nickase (see figures below). Cas9TX could efficiently eliminate chromatin abnormalities during genome editing, better than all the current editing toolboxes. Thirdly, we have performed PEM-seq analysis in various cell lines and human primary T cells to compare Cas9TX with Cas9 nuclease (**Figs.4 and 7**), indicating that Cas9TX could work at the editing scenarios compatible with Cas9. Fourthly, we also detected increased levels of small deletions (<100bp) as previously reported (**Fig. 3i** vs Allen et al., 2018, PMID: 30480667), indicating that TREX2 functioned consistently in our system.

2. technical: many experiments were done in a single replicate. this makes it impossible to know the distribution, and thereby creating reproducibility issues.

Thanks for the suggestions. We agreed with the reviewer that it's important to present reproducible experiments. In the last revised version, we have already generated a lot of data to make the conclusions more convincing. PEM-seq *per se* captures thousands of chromosomal translocations in each library, which improves the statistical accuracy. Moreover, we either performed PEM-seq analysis at multiple loci (≥ 3 enzymes or sites) to support our conclusions (**Figs. 1g, 2e, 2g, 3c, 5a-c, Supplementary Figs. 1g-h, 1k-l, 2f-2h, 3f-3g**) or generated three biological replicates for each locus to support some of the key conclusions (**Figs. 1b-e, 2f, 2h, 3d-f, 4b-c, 5b-g, 6b-g, Figs. 7, Supplementary Figs. 2c-d, 7a-g**).